# Regulation of symbiotic interactions and primitive lichen differentiation by *UMP1* MAP kinase in *Umbilicaria muhlenbergii*

Yanyan Wang [1,2], Rong Li[1], Diwen Wang [2], Ben Qian [1], Zhuyun Bian [2], Jiangchun Wei[1], Xinli Wei [1] ✉ & Jin-Rong Xu [2] ✉

Lichens are of great ecological importance but mechanisms regulating lichen symbiosis are not clear. *Umbilicaria muhlenbergii* is a lichen-forming fungus amenable to molecular manipulations and dimorphic. Here, we established conditions conducive to symbiotic interactions and lichen differentiation and showed the importance of *UMP1* MAP kinase in lichen development. In the initial biofilm-like symbiotic complexes, algal cells were interwoven with pseudohyphae covered with extracellular matrix. After longer incubation, fungal-algal complexes further differentiated into primitive lichen thalli with a melanized cortex-like and pseudoparenchyma-like tissues containing photo-active algal cells. Mutants deleted of *UMP1* were blocked in pseudohyphal growth and development of biofilm-like complexes and primitive lichens. Invasion of dividing mother cells that contributes to algal layer organization in lichens was not observed in the *ump1* mutant. Overall, these results showed regulatory roles of *UMP1* in symbiotic interactions and lichen development and suitability of *U. muhlenbergii* as a model for studying lichen symbiosis.

Lichen-forming fungi (mycobionts) account for approximately 20% of all known fungal species, and they form various lichen thalli with green algae or cyanobacteria (photobionts) in nature[1]. The symbiotic relationship between lichen-forming fungi and their photosynthetic partners is considered as a model of mutualism[2]. Besides having mutually beneficial and co-dependent metabolic states[3], the mycobiont and photobiont cells form stable lichens that are resistant against various environmental stresses and can thrive on harsh environments[4]. In lichens, the intimate interactions between fungal and photobiont cells involve the differentiation of specialized attachment or haustorium-like structures although their functions and importance are not clear[5]. To better understand mycobiont-photobiont symbiosis, it is important to characterize molecular mechanisms governing their interactions, establishment of initial symbiotic complexes, and development of primitive and mature lichen thalli. Unfortunately, most lichen-forming fungi are not amenable to molecular genetic studies, and artificial synthesis of

lichens with isolated mycobionts and photobionts is difficult due to their slow growth rate[6,7].

*Umbilicaria muhlenbergii* is a foliose lichen that grows on granitic rocks at high altitudes. To date, it is the only known lichen-forming fungus that undergoes dimorphic transition from yeast to hyphal/pseudohyphal growth when interacting with photobiont cells[8,9]. On potato dextrose agar (PDA) or other common fungal media, it grows relatively fast and normally takes only 7–10 days for a single yeast cell to form a visible colony. The photobiont partner for *U. muhlenbergii* is *Trebouxia jamesii*, a common green algal species in lichens[10–12]. Under laboratory conditions, the yeast-to-pseudohypha transition can be induced by nutrient starvation and treatments with extracellular cAMP or 3-isobutyl-1-methylxanthine (IBMX) in *U. muhlenbergii*, indicating the involvement of the cAMP-PKA pathway[9]. Pseudohyphal growth is also rapidly induced in *U. muhlenbergii* by algal cells of *T. jamesii*. After 10 days of co-cultivation on PDA, extensive pseudohyphal growth in close contact with algal cells could be observed[9]. However, these

[1]State Key Laboratory of Mycology, Institute of Microbiology, Chinese Academy of Sciences, Beijing 100101, China. [2]Dept. of Botany and Plant Pathology, Purdue University, West Lafayette, IN 47907, USA. ✉e-mail: weixl@im.ac.cn; jinrong@purdue.edu

fungal-algal cell clusters appeared to be defective in further development of symbiotic interactions or lichen thallus-like structures and tended to segregate into greenish (algal cells only) or whitish (yeast cells only) areas after longer incubation. The physiological and environmental conditions necessary to induce the establishment of stable symbiotic interactions between *U. muhlenbergii* and *T. jamesii* and further differentiation of lichen thalli in the lab remain to be investigated.

Interestingly, even on nutrient-rich medium, pseudohyphal growth of *U. muhlenbergii* can be induced by algal cells[9], suggesting that direct contacts with algae may activate other signaling pathways to trigger the yeast-to-pseudohypha transition besides the cAMP-PKA pathway. In the budding yeast *Saccharomyces cerevisiae*, pseudohyphal growth or filamentation is coregulated by multiple signaling pathways, including the cAMP signaling and mitogen-activated protein (MAP) kinase pathways[13,14]. Similarly, in the human pathogen *Candida albicans*, both cAMP signaling and MAP kinase (MAPK) pathways are involved in regulating the yeast-to-hypha transition, which is required for fungal pathogenesis[15,16]. In the corn smut fungus *Ustilago maydis*, the Kpp2/Kpp6 MAPKs and PKA also co-regulate plant infection processes, which involves dimorphic transition too[17,18]. In the rice blast fungus *Magnaporthe oryzae*, the *PMK1* MAP kinase, orthologous to yeast Fus3/Kss1, regulates late stages of appressorium formation and appressorial penetration, although cAMP signaling is responsible for surface recognition to initiate the formation of appressoria[19], which are highly specialized infection structures with distinct morphology from hyphae. Studies in other plant pathogenic fungi have showed that *PMK1* orthologs are essential for pathogenesis by regulating various infection-related morphogenesis and fungal-plant interactions in infected tissues[20,21]. However, the importance of this well-conserved MAP kinase pathway in fungal symbiosis is not clear. In fact, its orthologs have not been functionally characterized in any of the fungi that form symbiotic relationships with plants, animals, or algae.

Like all other lichens that are formed on hard surfaces, *U. muhlenbergii* develop thalli on rocks in nature. To further characterize the mycobiont-photobiont interaction under conditions mimicking its natural environments, in this study, we showed that co-cultivation of *U. muhlenbergii* and *T. jamesii* on glass or dialysis membrane stimulated the development of initial fungal-algal symbiotic complexes in which algal cells were dispersed and interwoven by capsulated pseudohyphae. These biofilm-like fungal-algal complexes could further differentiate into primitive lichen thalli that had a highly differentiated and melanized cortex-like layer and dispersed algal cells in pseudoparenchyma-like tissues. Inhibition or deletion of the *UMP1* MAP kinase blocked pseudohyphal growth of *U. muhlenbergii* and its ability to establish symbiotic interactions with algal cells. Overall, our data showed that *U. muhlenbergii* can form initial symbiotic complexes with algal cells and further differentiate into primitive lichen thalli on artificial substrates. This differentiation process, similar to fungal pathogenesis, is regulated by the *UMP1* MAP kinase pathway. The differentiation of primitive lichen thalli and characterization of signaling pathways regulating symbiotic interactions make *U. muhlenbergii* a great model for studying molecular mechanisms of lichen symbiosis or fungal symbiosis in general.

## Results

### Formation of biofilm-like symbiotic complexes by fungal-algal cells attached to glass

In a previous study, we observed initial fungal-algal interactions but not further differentiation on potato dextrose agar (PDA)[9]. As *U. muhlenbergii* lichens are formed on rocks in nature, we first tested the attachment of fungal-algal cells to glass. After incubation of a thin layer (2 mm) of 0.1× potato dextrose broth (PDB) cultures of mixed mycobiont and photobiont cells (1:10) for 10 days in glass flasks, a greenish layer of fungal-algal cells with granular appearance was formed on the bottom, which could not be dislodged by vigorous shaking (Fig. S1a). This was not observed in the controls with the mycobiont and photobiont cells cultured separately.

Due to difficulties with microscopically examination of fungal-algal cells attached to the bottom of glass flasks, we submerged a sterilized slide glass in a petri plate with the same thin layer of 0.1×PDB cultures of fungal-algal cells. After incubation for 10 days, a similar layer of greenish, granular cell masses was observed on the slide glass, which was resistance against gentle rinsing and stained purplish with crystal violet (Fig. S1b), a dye used to stain bacterial biofilms[22]. Microscopic examination of fungal-algal cell masses attached to the slide glass after rinsing with distilled water showed that more than 90% of the mycobiont grew as pseudohyphae in association with algal cells in three independent replicates, with 300 fungal cells examined in each replicate (Fig. 1a). Although some algal cells appeared to be only loosely associated with fungal cells, some were surrounded with interwoven pseudohyphae (Fig. 1b). Some areas of fungal-algal association complexes were darkly pigmented (Fig. 1b), which is likely due to melanin production by the mycobiont.

### Observation of the capsule layer of pseudohyphae and extracellular matrix in symbiotic complexes

Since submerged cultures are not exposed to the air, we then tested with sterile dialysis bags made of regenerated cellulose membranes (hereafter referred to as cellulose membrane) placed over the surface of 0.1×PDA. Drops of the fungal-algal mixture (1:10) were gently spread over cellulose membranes. After incubation under light for 10 days, the fungal-algal association complexes formed on cellulose membranes appeared to be lumpy and had spreading pseudohyphae with mucinous appearance (Fig. 1c). Negative staining with Indian ink[23] showed that these pseudohyphae had a capsule layer (Fig. 1d). In yeast cells of *U. muhlenbergii* cultured without algae, such a capsule layer was not observed (Fig. 1d).

When examined by cryo-scanning electron microscopy (cryo-SEM), we examined 50 fungal-algal complexes formed on cellulose membranes and found that almost 100% of these complexes were covered with extracellular matrix although there were areas with visible superficial or protruding pseudohypha/hyphae (Fig. 1e). Close examination showed that pseudohyphae on the surface and algal cells were connected with mucous fimbriae or covered with extracellular matrix (Fig. 1f). At the edge of fungal-algal complexes, the extracellular matrix spread from cell masses over to cellulose membranes (Fig. 1g), which may enhance surface attachment. When the same samples were examined by transmission electron microscopy (TEM), fungal cells but not algal cells were often enveloped in a relatively thick capsule layer (Fig. 1h), suggesting that the extracellular matrix may be mainly produced by the mycobiont.

### Invasion of dividing algal mother cells by the mycobiont

When cultured alone, algal cells of *T. jamesii* normally aggregate into clusters due to the multiple fission nature of its cell division. However, fungal cells tended to disperse and intertwine with algal cells in the fungal-algal symbiotic complex formed on glass or cellulose membranes (Fig. 2a). When examined by SEM, pseudohyphae were often observed to grow among daughter cells that were still covered by the outer cell wall of mother cells (Fig. 2b). In TEM examinations, the mycobiont was also frequently observed in the middle of dividing mother cells (with visible daughter cells) (Fig. 2c). These results indicate that *U. muhlenbergii* has the ability to penetrate or invade dividing mother cells that may be compromised in cell wall integrity during division into multiple daughter cells.

To investigate whether invasion of dividing algal mother cells[24,25] is a natural occurrence in *U. muhlenbergii* lichens, we examined lichen thalli freshly collected from rocks. In the algal zone situated between the upper cortex and medulla layer, algal cells were intwined with

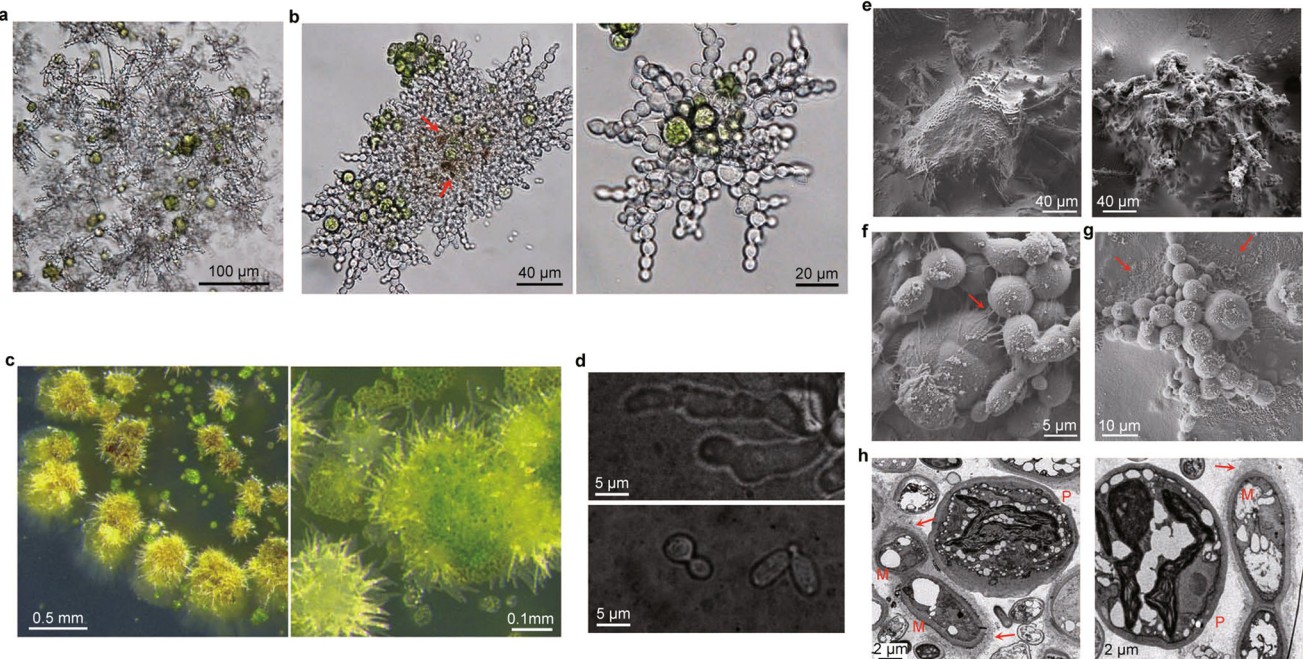

**Fig. 1 | Formation of the fungal-algal complex on glass slide and cellulose membranes. a** Fungal-algal masses adhered to glass slide after co-cultivation of *U. muhlenbergii* and *T. jamesii* cells (1:10) for 10 days in 0.1×PBD. Pseudohyphae (the status of mycobiont in association with algal cells, encompassing 90% of the total mycobiont here), dispersed algal cells, and accumulation of mucinous materials surrounding fungal-algal cells were observed. Five (*n* = 5) independent experiments were conducted, with no significant differences between replicates. **b** Close-up view of a fungal-algal cell cluster. Most algal cells were dispersed and interwoven with pseudohyphae. Red arrows marks area with melanization. **c** Fungal-algal complexes formed on cellulose membranes after co-incubation for 10 days, with spreading/protruding pseudohyphae on the edge or surface. Same results in three (*n* = 3) independent experiments. **d** Yeast cells of *U. muhlenbergii* (lower) and pseudohyphae induced by algal cells on cellulose membranes (upper) were stained with India ink. Only pseudohyphae have the capsule layer. **e** The surface of fungal-algal cell masses formed on cellulose membranes were examined by Cryo-SEM. A total of fifty cell masses were observed and almost 100% of the complexes had extracellular matrix. **f, g** Close observation of superficial pseudohyphae and algal cells that were connected or covered with extracellular matrix. Arrows point to mucinous fimbriae over algal cells or extracellular matrix spreading from the fungal-algal complex to cellulose membranes. **h** Examination of the fungal-algal complex by transmission electron microscopy (TEM). The mycobiont cells (M) in associated with photobiont cells (P) had a relatively thick capsule layer (marked with arrows). More than one hundred fungal cells were observed and over 90% had thick capsule layer.

hyphae of the mycobiont (Fig. 2d). In cross-sections of *U. muhlenbergii* lichen thalli, fungal growth among daughter cells inside dividing mother cells was frequently observed by cryo-SEM examinations (Fig. 2e). Growth of the mycobiont inside dividing algal mother cells was also observed by TEM examination (Fig. 2f). There, invasion of dividing mother cells by the mycobiont occurs in natural *U. muhlenbergii* lichens, which may contribute to the entanglement and dispersal of algal cells in the algal zone. Interestingly, fungal cells in close contact with algal cells occasionally underwent morphological differentiations and formed structures (Fig. 2g) that were similar to what have been described as appressoria and haustoria in other lichen-forming fungi[5,26]. Similar to fungal-algal symbiotic complexes formed on cellulose membranes, fungal cells, but not algal cells, were surrounded with a relatively thick capsule layer in the lichen thallus (Fig. 2f, g).

## Primitive lichen thalli formed on cellulose membranes have similar features with natural lichens

To assay for further differentiation of symbiotic complexes formed on cellulose membrane, we kept the co-cultivation plates in a light incubator with a relative humidity of 75% for an extended period of time. After incubating for three months, fungal-algal co-cultures with varying degrees of melanization were observed on the dried 0.1×PDA and cellulose membrane (Fig. 3a). At least 90% of the symbiotic complexes had a hairy appearance with protruding or extending hyphae/pseudohyphae on the surface or at the edge. Such kind of differentiation was not observed in 3-month-old mycobiont cells cultivated alone (Fig. S2a). To test whether algal cells were still alive, we added 1 ml sterile water to each plate to rehydrate the co-cultivation samples.

After incubation for another week, these lichen thallus-like structures (referred as primitive lichen thalli below) became watery in appearance, likely due to rehydration of the extracellular matrix, and had visible growth at the edge (Fig. 3b).

Microscopical examinations showed that the cortex-like layer of primitive lichen thalli was composed of melanized and highly differentiated fungal cells, and some of them had hypha-like filaments on the surface (Fig. 3c), which is similar to the upper cortex of natural lichens. The algal cells released from crashed primitive lichen thalli were greenish and exhibited red chlorophyll autofluorescence (Fig. 3c). When stained with fluorescein diacetate (FDA), 75.3 ± 2.0% of these algal cells had green fluorescence signals that overlapped with chlorophyll autofluorescence (Fig. 3d, Table S1). In the control with the photobiont cultured alone, 92.3 ± 0.9% of the algal cells were grayish and shriveled, lacked chloroplast fluorescence, and could not be stained with FDA (Fig. S2b, Table S1). These results indicate that the algal cells trapped in primitive lichen thalli were alive and likely retained photosynthesis capacities.

When examined by cryo-SEM, we observed that many primitive lichen thalli had a downward growth and pushed down the cellulose membrane in the center (Fig. 3e), which may be similar to early stages of umbilicus differentiation in *U. muhlenbergii* lichens. These primitive lichen thalli had protruding pseudohyphae/hyphae on the surface (Fig. 3e) and a cortex-like layer consisted of highly differentiated fungal cells (Fig. 3f, g). The algal cells were mainly found underneath the cortex-like layer and surrounded/embedded by interwoven pseudohyphae or hyphae and extracellular matrix (Fig. 3f, g), which is structurally similar to pseudoparenchyma of natural lichens. It appears that

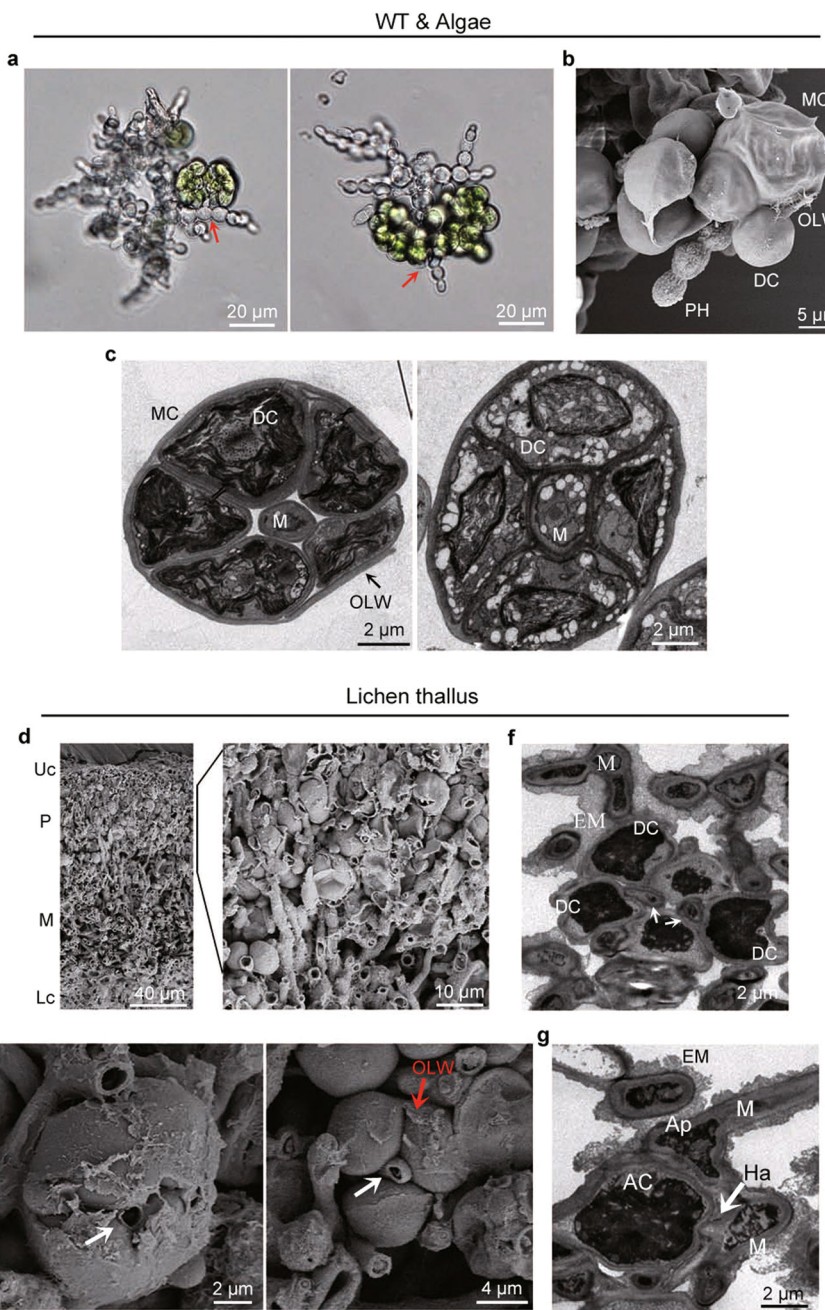

**Fig. 2 | Fungal-algal interactions in the symbiotic association complex or lichen thallus. a** Pseudohyphal growth among algal cells observed in the fungal-algal complex formed on cellulose membranes after co-incubation for 10 days. Arrows point to the sites pseudohyphae entering or emerging from aggregates of algal cells. **b** A dividing mother cell (MC) with a pseudohypha (PH) observed by SEM examination. **c** A dividing mother cell with a mycobiont cell (M) in the center was observed by TEM. The co-culture was subjected to three ($n = 3$) independent experiments with similar phenotypes in each culture plate. **d** Cross-section of a mature *U. muhlenbergii* lichen thallus examined by SEM. The algal zone was amplified to show algal cells interwoven with fungal cells. Uc, upper cortex; P, algal zone; M, medulla; Lc, lower cortex. **e** Close examination of the algal zone by SEM to show dividing mother cells invaded by mycobiont cells (marked with arrows). **f** TEM examination of a dividing mother cell with daughter cells (DC) and a mycobiont cell in the middle (marked with arrows). **g** The appressorium-like structure (Ap) developed at the deformed, swollen hyphal tip and a haustorium-like structure (Ha) defined as a protrusion into algal cells (AC) by the mycobiont (M). For F and G, the mycobiont cells had a relatively thick layer of extracellular matrix (EM). For B, C, and E, OLW stands for out layer wall of the mother cell. Three ($n = 3$) individual thallus were examined, all with similar phenotypes.

the photobiont is protected by the mycobiont against desiccation because most of the algal cells were not collapsed. Furthermore, we observed the invasion of dividing algal mother cells by the mycobiont and the formation of extracellular matrix in these primitive lichen thalli by TEM examinations (Fig. 3h). Taken together, these results indicate that the lichen thallus-like structures formed on cellulose membranes have many structural and functional features that are similar to those of natural lichen thalli.

## Treatment with U0126 inhibits pseudohyphal growth and symbiotic interactions with algal cells

Because of the important roles of MAP kinase pathways in pathogenesis and fungal-plant interactions in plant pathogens[20,21], we treated fungal-algal mixtures with MAPK kinase (MEK) inhibitors PD98059 and U0126[27]. In liquid cultures, treatments with 30 μM U0126 resulted in a significant reduction in the amount of fungal-algal cell masses adhered to the submerged glass slide (Fig. S3a). Instead of a greenish, granular

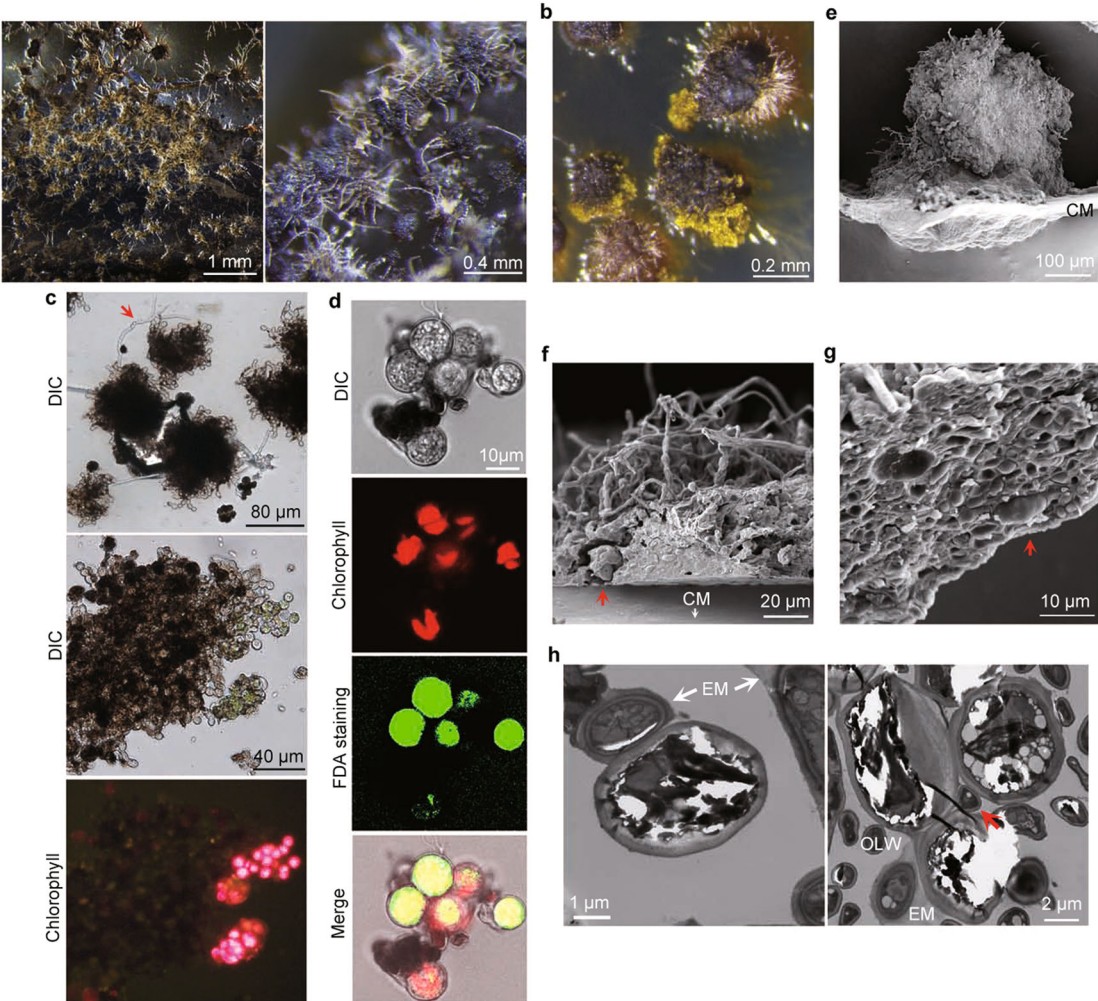

**Fig. 3 | Formation of primitive lichen thalli on cellulose membranes. a** Lichen thallus-like structures formed by co-cultivation of fungal-algal cells on cellulose membranes over a thin, desiccated layer of 0.1×PDA for 3-month. Arrows mark hyphae/pseudohyphae on the surface. Plates from three independent experiments (*n* = 3) were examined and all had similar results. **b** Close examination of individual primitive lichen thalli after wetting with sterile distilled water for a week. Growth from the edge was marked with arrows. **c** Crushed primitive lichen thalli were examined for the heavily melanized cortex-like layer with protruding hyphae/pseudohyphae (top), highly differentiated fungal cells and green algal cells (middle), and chlorophyll autofluorescence (bottom). **d** Algal cells released from crashed primary lichens were stained with fluorescein diacetate (FDA) and examined by DIC and epifluorescence microscopy. 75.3 ± 2.0% of algal cells examined had FDA staining signals overlapping with chlorophyll autofluorescence. The statistical data are presented in Supplementary Table 1. **e** Side view of a primitive lichen thallus examined by cryo-SEM. Downward growth of the fungal-algal cells pushed down the cellulose membrane (CM). Arrows mark the hyphae or pseudohyphae protruding out from the surface. Five primitive lichen thalli were observed laterally, all with similar phenotypes. **f**, **g** Cross sections of primitive lichen thalli were examined by Cryo-SEM to show the cortex-like layer of fungal cells above the pseudoparenchyma-like tissues consisting of pseudohyphae and algal cells (marked with arrows). More than 20 primitive lichen thalli were longitudinally, all with similar phenotypes. CM, the surface of cellulose membranes. **h** Extracellular matrix (EM) formed by the mycobiont (left) and a dividing mother cell and its daughter cells with a mycobiont cell (marked with an arrow) in the middle (right) were observed in a cross-section of primitive lichen thalli examined by TEM. OLW, out layer wall of the mother cell.

layer observed in the untreated control, only a few spots of fungal-algal cells were observed in the U0126-treated samples after gentle rinsing. Treatment with 40 μM PD98059 also reduced the attachment of fungal-algal cells to slide glass, but to a lesser degree compared to U0126 (Fig. S3a).

We then repeated this experiment with cellulose membranes and obtained similar results. Treatments with PD98059 and U0126 reduced the formation and growth of fungal-algal symbiotic complexes (Fig. S3b). Microscopical examination showed that only limited, short pseudohyphae were developed by the mycobiont in the presence of PD98059 (Fig. 4a). However, U0126 was more effective in inhibiting pseudohyphal growth and fungal-algal symbiotic interactions. Pseudohyphal growth was blocked by U0126 in *U. muhlenbergii* co-cultivated with algal cells (Fig. 4a). These results indicated that U0126 is more effective than PD98059 in inhibiting

pseudohyphal growth and symbiotic interactions with algal cells in *U. muhlenbergii*.

### Deletion of *UMP1* results in defects in pseudohyphal growth

Like many other ascomycetes, *U. muhlenbergii* has three typical MAPK genes, including Um05c1700 that is homologous to *PMK1* and named *UMP1* (for *U. muhlenbergii PMK1*) in this study (Figs. S4, S5). Because of the conserved functions of *PMK1* orthologs in regulating infection-related morphogenesis and fungal-plant interactions[19,20], we used the split-marker approach to generate mutants deleted of *UMP1* (Fig. S6) to investigate its role in fungal-algal interactions. The *ump1* deletion mutant displayed normal yeast cell morphology and growth rate (Fig. S6). However, unlike the wild type[9], the *ump1* mutant was defective in the yeast-to-pseudohypha transition induced by osmotic stress or nutrient starvation (Fig. 4b). Treatments with IBMX, an inhibitor of

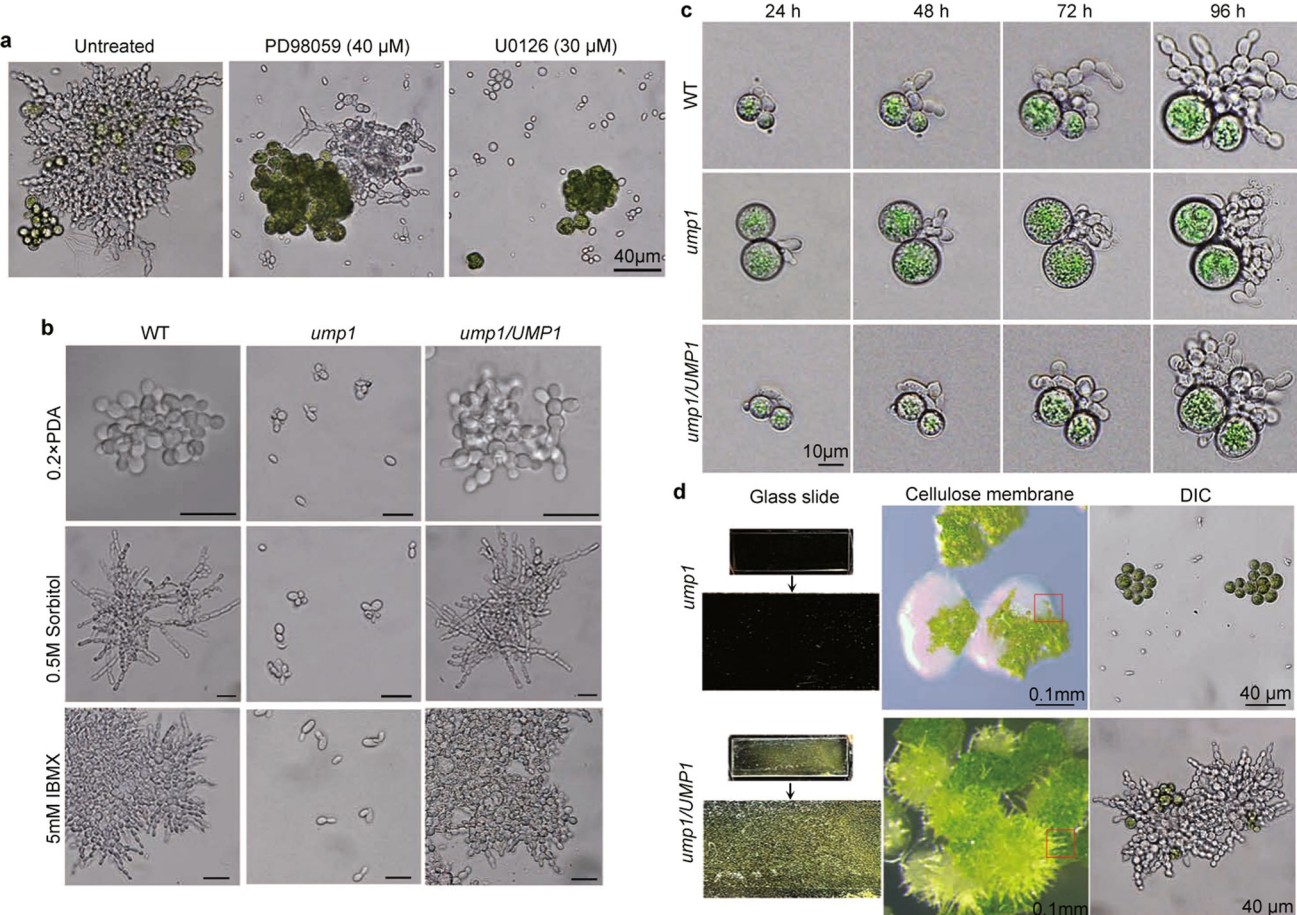

**Fig. 4 | Effects of MEK inhibitors and UMP1 deletion on fungal-algal interactions and pseudohyphal growth. a** Slide glass submerged in the 0.1×PDB cultures of fungal and algal cells (1:10) with or without labeled MEK inhibitors for 10 days were gently rinsed and examined. U0126 was more effective than PD98059 in inhibiting the formation of fungal-algal symbiotic complex attached to slide glass. Untreated and treated co-culture samples were subjected to three independent experiments ($n = 3$) and all replicates had similar results. **b** Ten-day-old cultures of the wild type (WT), *ump1* mutant, and *ump1/UMP1* complementation transformant cultured on 0.2×PDA (nutrient starvation) and PDA with 0.5 M sorbitol (osmotic stress) or 5 mM IBMX were examined for pseudohyphal growth. Pseudohyphae were not observed in the *ump1* mutant. Bar = 20 μm. Three independent experiments ($n = 3$) under each treatment condition had similar results. **c** Co-cultures of labeled strains with algal cells were examined for pseudohyphal growth every 24 h (up to 96 h). Pseudohyphal growth was induced by algal cells in the wild type and complementation transformant but not in the *ump1* mutant. For each group of samples, we observed ten fungal-algal pairs ($n = 10$) all with similar results. **d** Co-cultures of algal cells with the *ump1* mutant or *ump1/UMP1* complementation transformant on slide glass and cellulose membranes. Microscopical examination showed that the *ump1* mutant grew only as yeast cells even from areas associated with lumps of algal cells. Red squares mark the areas sampled for DIC microscopical examination. Three independent experiments ($n = 3$) were conducted in co-culture, and representative results are presented.

cAMP diphosphoesterase, also failed to induce pseudohyphal growth in the *ump1* mutant (Fig. 4b). In co-cultivation assays, the *ump1* mutant showed no pseudohyphal growth in yeast cells associated with algal cells (Fig. 4c).

For complementation assays, we generated the *UMP1*-GFP fusion construct under the control of its native promoter and transformed it into the *ump1* mutant. The *ump1/UMP1*-GFP transformant was normal in pseudohyphal growth under osmotic or starvation conditions and IBMX treatments (Fig. 4b). Co-cultivation with algal cells also induced the yeast-pseudohypha transition in the *ump1/UMP1*-GFP transformant (Fig. 4c), indicating that expression of *UMP1*-GFP fully complemented the *ump1* mutant. Therefore, the Gpmk1 MAPK pathway plays a critical role in regulating pseudohyphal growth in *U. muhlenbergii*.

### *UMP1* is essential for the formation of fungal-algal complexes and invasion of dividing mother cells

To determine the effect of *UMP1* deletion on symbiotic interactions, cells of *ump1* mutant and *ump1/UMP1*-GFP strains were co-incubated with algal cells for 10 days, as described above. Pseudohyphae and fungal-algal cell masses adhered to submerged slide glass were not

observed in the *ump1* mutant after gentle rinsing (Fig. 4d). On cellulose membranes, the *ump1* mutant grew as yeast cells whether in direct contact with algal cells or not (Fig. 4d). Under the same conditions, like the wild type, the *ump1/UMP1*-GFP strain showed normal pseudohyphal growth and formed symbiotic complexes with algal cells.

When examined by SEM, algal cells mainly existed in clusters without intertwining pseudohyphae, although there were a few yeast cells on the surface in the 10-day-old *ump1*-algal cell masses (Fig. 5a). Under the same conditions, algal cells co-cultivated with the *ump1/UMP1* strain were embedded in pseudohyphae (Fig. 5a). Close examination showed that the *ump1* mutant was defective in invading dividing mother cells of *T. jamesii* (Fig. 5a). Pseudohyphal growth among algal cells was only observed in the *ump1/UMP1*-GFP strain. These results indicate that *UMP1* is essential for *U. muhlenbergii* to invade dividing mother cells.

### The activation of Ump1 and its localization to the nucleus are stimulated by algal cells

To further determine the role of Ump1 in fungal-algal interactions, we assayed its expression and phosphorylation with an anti-Gpmk1

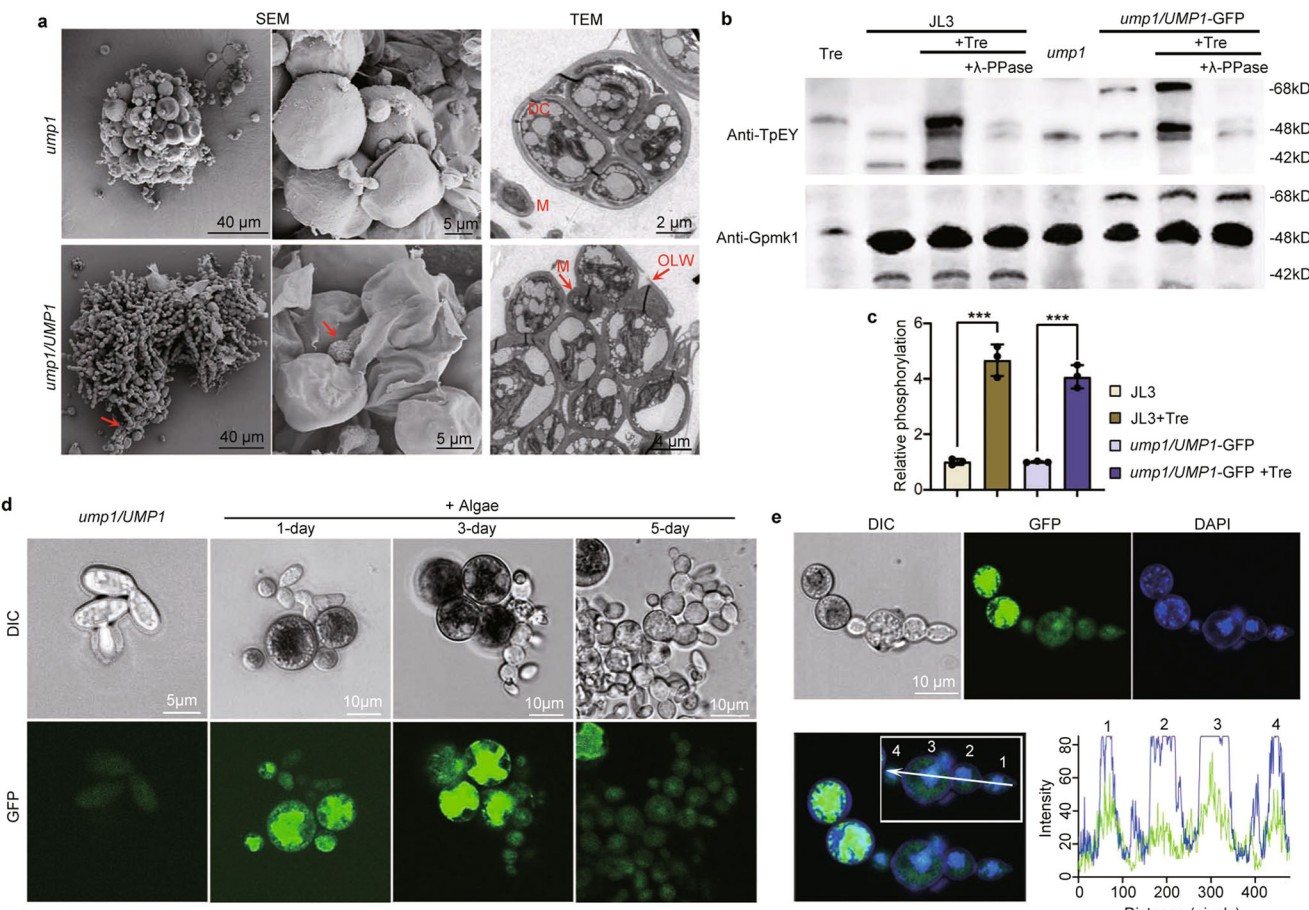

**Fig. 5 | Essential roles of UMP1 in symbiotic interactions with algal cells. a** SEM and TEM examination of algal cells co-cultured with the *ump1* mutant or *ump1/ UMP1* complementation transformant on cellulose membranes for 10 days. Arrows point to pseudohyphae associated with algal cells. DC, daughter cells; M, mycobiont cells; OLW, out layer wall of algal mother cells. **b** Western blots of proteins isolated from fungal-algal cocultures and algal or fungal cells were detected with the anti-TpEY and anti-Gpmk1 antibodies. Contacts with algae significantly increased the phosphorylation level of Ump1 (42-kD) and Ump1-GFP fusion protein (68-kD). Phosphorylation of Ump1 and Ump1-GFP disappeared after treatment with λ- phosphatase. The anti-TpEY antibody but not anti-Gpmk1 antibody detected one MAP kinase in algal cells. In fungal cells, the 48-kD UmMps1 band was detected with both anti-Gpmk1 and anti-TpEY antibodies. **c** Relative phosphorylation level of Ump1 and Ump1-GFP fusion protein in fungal cells cultured alone (arbitrarily set to 1) or with algae (+Tre). Intensities of the 42-kD Ump1 band and

the 68-kD band were quantified with the Image-Pro Plus software to estimate the ratio of band intensities detected with the anti-TpEY and anti-Gpmk1 antibodies. Mean and standard deviation were estimated with data from three independent replicates ($n = 3$). ***$p = 0.0004$ (two-tailed Student's *t* test). **d** Cells of the *ump1/UMP1* transformant cultured alone (far left) or with algae after incubation for 1, 3, and 5 days were examined by DIC and epifluorescence microscopy. Contacts with algal cells had no or only a minor effect on GFP signals in the cytoplasm but increased GFP signals in the nucleus, particular after co-cultivation for three days. **e** Co-localization analysis of GFP and DAPI signals in the nucleus in pseudohyphae formed by the *ump1/UMP1*-GFP transformant associated with algal cells after co-cultivation for three days. The white arrow in the high-magnification panel marks the four fungal cells used to generate the line scan graph on the right, which showed that the peaks of fluorescence signals of Ump1-GFP (green) and DAPI staining (blue) overlap in all four fungal cells examined.

antibody and an anti-phospho-p44/42 MAPK antibody that specifically detects phosphorylation at the TEY dual-phosphorylation sites. On western blots with total proteins isolated from fungal or algal cells cultured separately and 6 h fungal-algal cocultures on cellulose membrane, the 42-kD Ump1 band of *U. muhlenbergii* was detected in the wild type but not in the *ump1* mutant by both antibodies (Fig. 5b), confirmed the deletion of *UMP1* and detection of Ump1 proteins. In the *ump1/UMP1*-GFP complementation strain, the 68-kD Ump1-GFP fusion band was detected with anti-Gpmk1 and anti-GFP (Fig. 5b; S7) antibodies. The expression of Ump1 was not but its phosphorylation was significantly increased in the fungal-algal co-cultures compared to fungal cells cultured alone (Fig. 5b). When treated with λ-phosphatase, the 42-kd Ump1 band was still detected by the anti-Gpmk1 antibody but no longer detectable by the anti-phospho-p44/42 MAPK antibody (Fig. 5b). Phosphorylation of the Ump1-GFP fusion protein was also increased after co-cultivation with algae (Fig. 5b). When quantified, the relative phosphorylation level of Ump1 increased 4-fold in fungal cells co-cultivated with algae (Fig. 5c). *U. muhlenbergii* has another MAP

kinase with the TEY dual phosphorylation sites that is homologous to Mps1 of *M. oryzae*[28]. Although the expression of 48-kD UmMps1 was not increased by co-cultivation with algal cell, we had difficulty assaying its phosphorylation level due to the detection of an algal MAPK band of the similar size by the anti-phospho-p44/42 MAPK antibody (Fig. 5b).

Since fungal MAP kinases are known to have dynamic localization during differentiation[29], we examined GFP signals in the *ump1/UMP1*-GFP strain. Without algal cells, only weak GFP signals were observed in the cytoplasm of yeast cells (Fig. 5d). When co-cultivated with algae, GFP signals in the cytoplasm of fungal cells were not affected. However, GFP signals in the nucleus of pseudohyphae were increased by co-cultivation with algal cells, especially after incubation for three days (Fig. 5d). To confirm the localization of Ump1-GFP fusion proteins to the nucleus, we stained fungal-algal cells with DAPI. The GFP signals overlapped with the DAPI staining signals in the pseudohyphae induced by algae (Fig. 5e), indicating that localization of Ump1-GFP proteins to the nucleus is stimulated by interactions with algal cells.

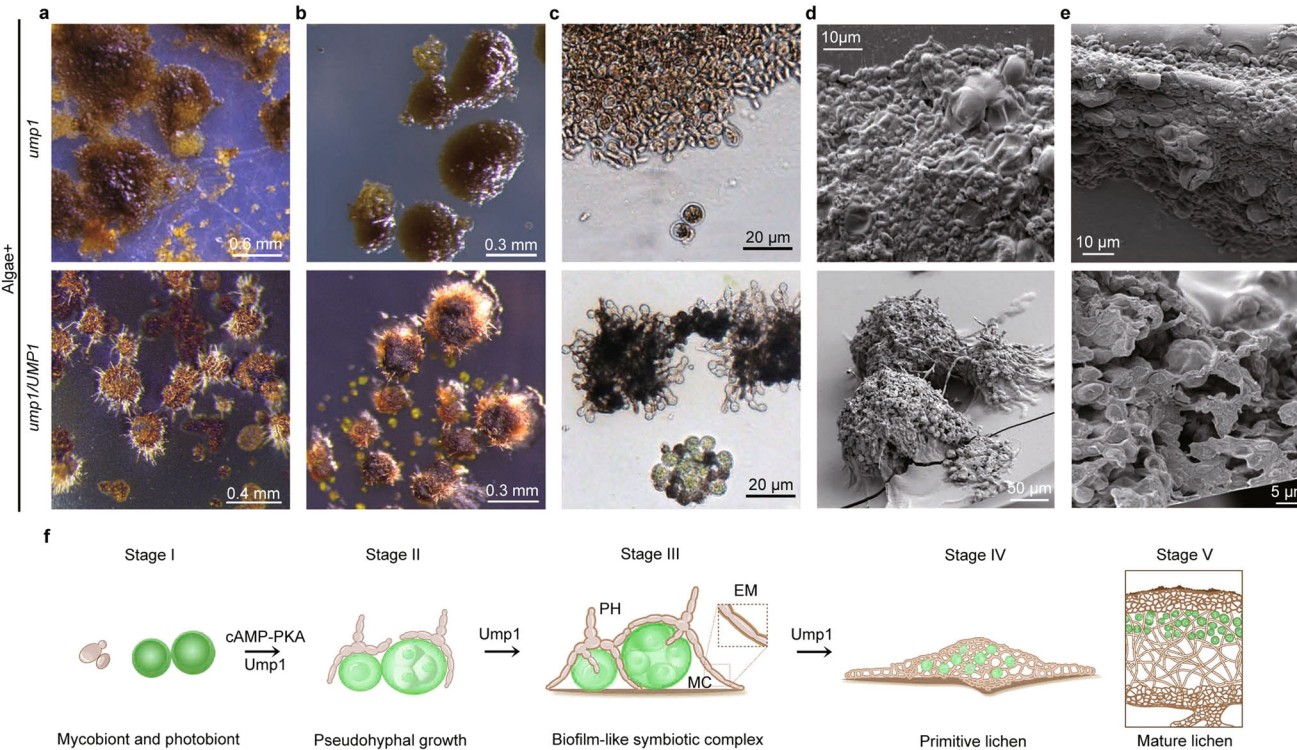

**Fig. 6 | Defects of the ump1 mutant in the formation of primary lichen thalli.**
**a** Co-cultivation of algal cells with the *ump1* mutant or *ump1/UMP1* complementation transformant on cellulose membranes for three months. The *ump1* mutant had slight melanization of fungal cells but failed to form lichen thallus-like structures. **b** The morphology of the same set of 3-month-old fungal-algal cocultures after being wetted for a week with sterile water. Whereas primitive lichen thalli formed by the *ump1/UMP1* strain grew on the edge, no further growth was observed in the *ump1* mutant. **c** Microscopical examination of fungal and algal cells in 3-month-old cocultures. No pseudohyphae or highly differentiated fungal cells were observed and most algal cells were not green in the *ump1* mutant. **d** Differences in the surface appearance between *ump1*-algal cell masses and *ump1/UMP1* primitive lichen thalli examined by cryo-SEM. The *ump1*-algal cell masses had collapsed cells and lacked pseudohyphae/hyphae and extracellular matrix. **e** Cross sections of *ump1*-algal cell masses and *ump1/UMP1* primitive lichen thalli examined by cryo-SEM. Pseudoparenchyma-like structures consisting of pseudohyphae and algal cells were

only observed in the latter. **f** Diagrams of fungal-algal interactions at different stages of lichen synthesis. Contacts with algal cells induce pseudohyphal growth (II). Attachment to a hard surface stimulates further differentiation of biofilm-like symbiotic fungal-algal complexes (III) with the penetration of dividing mother cells (MC) by pseudohyphae (PH) and formation of extracellular matrix (EM), which may extend over the edge onto cellulose membranes. Primitive lichen thalli (IV) are developed after longer co-cultivation. Under the highly differentiated cortex-like layer with protruding hyphae/pseudohyphae, primitive lichen thalli contain photoactive algal cells interwoven with pseudohyphae. The *UMP1* MAPK pathway is essential for stages II-IV differentiation under laboratory conditions but the formation of mature lichens (V) thus far is only observed in nature. The cAMP-PKA pathway is involved in pseudohyphal growth but its role in late stages of differentiation and relationship with *UMP1* remain to be characterized. For **a**–**c**, plates from three independent experiments (*n* = 3) were examined and all had similar results. For **d**–**e**, fifty cell masses were observed, all had similar results.

## The *ump1* mutant failed to form primitive lichen thalli

We also examined 3-month-old co-cultures of algal and *ump1* cells on cellulose membranes. Unlike the wild type or *ump1/UMP1*-GFP strain, the *ump1* mutant failed to form hairy primitive lichen thalli (Fig. 6a). The surface of *ump1*-algal cell masses was relatively smooth and had lighter pigmentation than primitive lichen thalli formed by the *ump1/UMP1*-GFP strain (Fig. 6a). After addition of sterile water and incubation for a week, the *ump1*-algal cell masses had no obvious growth (Fig. 6b). Under the same conditions, growth at the edge of the primitive lichen thalli was observed in the *ump1/UMP1*-GFP strain. Microscopic observation showed that algal cells in 3-month-old *ump1*-algal cell masses appeared to be dead because greenish algal cells were only observed in primitive lichen thalli of *ump1/UMP1*-GFP (Fig. 6c).

When examined by cryo-SEM, no pseudohyphae were observed on the surface and most algal cells appeared to be collapsed in 3-month-old *ump1*-algal cell masses (Fig. 6d). Under the same conditions, pseudohyphae and extracellular matrix were observed on the surface of primitive lichen thalli formed by the *ump1/UMP1*-GFP strain (Fig. 6d). SEM examinations of cross sections showed that primitive lichen thalli formed by the *ump1/UMP1*-GFP strain had pseudoparenchyma-like tissues with algal cells intertwined with

pseudohyphae, which was not observed in the *ump1*-algal cell masses (Fig. 6e). These results indicate that *UMP1* is essential for symbiotic interactions with algae and differentiation of primitive lichen thalli.

## Discussion

As the only known dimorphic lichen forming fungus, pseudohyphal growth can be induced in *U. muhlenbergii* by IBMX and algal cells of its photobiont on PDA plates. In this study, we showed that fungal-algal cells adhered to glass or cellulose membranes formed a biofilm-like symbiotic complex at early stages and further differentiated into lichen thallus-like structures in 3-month-old, desiccated co-cultures. In 10-day-old symbiotic complexes, algal cells were often entangled or interwoven with pseudohyphae of *U. muhlenbergii* that had a capsule layer. Recently, lichens have been compared to microbial biofilms, particularly fungal biofilms, for their similarities in physiological traits and growth patterns as communities attached to a surface[30]. We described these fungal-algal symbiotic complexes as biofilm-like because they attached tightly to the glass or cellulose membranes and had an extracellular matrix layer. In *Candida albicans*, the first stage in biofilm formation is the adherence of yeast cells to the substrate[31]. Although its components are not clear, the capsule layer is unique to pseudohyphae and likely important for surface attachment

and interactions with algal cells in the biofilm-like symbiotic complexes. In *Cryptococcus neoformans*, capsule formation is important for stress response and pathogenesis[32]. In *U. muhlenbergii*, the capsule layer of pseudohyphae may play a critical role in symbiosis by providing protections against desiccation and other environmental stresses. Consistent with this hypothesis, we found that mycobiont cells in the primitive or natural lichen thalli also had a capsule layer, which is similar to reported extra hyphal, gel-like matrix in *U. muhlenbergii*[24] and extracellular matrix in other lichens[33,34], suggesting that it is a common phenomenon in lichen-forming fungi.

Several observations made us to describe the 3-month-old, desiccated fungal-algal cell masses as primitive lichen thalli or lichen-like structures in this study. First, their heavily melanized cortex-like layer consisted of highly differentiated fungal cells, which is similar to the upper cortex of *U. muhlenbergii* lichens[35]. Secondly, beneath the cortex-like layer, algal cells were loosely interspersed and entangled with pseudohyphae, which is similar to pseudoparenchyma tissues in natural lichens[36]. Another evidence is the protection of algal cells by the mycobiont against desiccation. Majority of algal cells in these primitive lichen thalli were green and alive with chloroplasts after three months. Addition of water could revive fungal and algal growth as well reactivate photosynthetic activities. Furthermore, although primitive lichen thalli were small, their overall appearance was similar to lichens when observed under a stereoscope. We also observed downward growth in the center of these primitive lichen thalli, which may resemble umbilicus differentiation by *U. muhlenbergii* lichens in nature although it remains possible that downward growth in desiccated cultures is related to water or nutrient stress. Resynthesis of lichen thalli with isolated mycobiont and photobiont cells has been reported for a number of lichens after developing cultivation conditions stimulatory to fungal-algal symbiosis, including *Acarospora fuscata* and *Cladonia cristatella*[37–43]. It is tempting to speculate that longer incubation time and improved culture conditions (such as dry-wet cycles and temperature shifts) as well as attachment to rocks from natural habitats (with the native microbiome on the surface) may lead to the differentiation of mature lichen thalli by *U. muhlenbergii* in the lab.

Like many other green algae, the mother cell of *T. jamesii* grows by multiple fission[43], leading to the formation of aggregated cell masses in cultures. However, algal cells are dispersed and interwoven with fungal hyphae in natural lichens[24,44,45] although the underlying mechanism is not clear. In this study we showed that pseudohyphae of *U. muhlenbergii* could invade dividing mother cells as early as in initial biofilm-like symbiotic complexes. Similar invasion of dividing mother cells and growth of pseudohyphae among daughter cells were observed in primitive and natural lichen thalli, which allows the mycobiont to disperse and entangle algal cells with a network of pseudohyphae. Therefore, it is possible that invasion of dividing mother cells is one of the mechanisms used by *U. muhlenbergii* and other lichen forming fungi to construct the algal layer in natural lichens. Nevertheless, penetration of an algal cell will be detrimental to the photobiont. To maintain symbiotic associations with algal cells, the mycobiont must be able to specifically recognize and penetrate dividing mother cells. The *UMP1* MAP kinase pathway may be involved in regulating these processes because the *ump1* mutant was defective in invasion of dividing mother cells and *UMP1* orthologs are important for plant penetration[46,47].

In *U. muhlenbergii*, an earlier study showed that the cAMP-PKA pathway is involved in regulating pseudohyphal growth[9]. Treatments with IBMX induces pseudohyphal growth in the wild type and *Umgpa3* deletion mutant. In this study, *UMP1* was found to be essential for the yeast-to-pseudohypha transition. Pseudohyphal growth could not be induced in the *ump1* deletion mutant by stresses, cocultivation with algal cells, or IBMX. In human and plant pathogenic fungi, this MAPK pathway and cAMP signaling often co-regulate various infection

processes[20,21]. For example, whereas Pmk1 regulates appressorium formation and penetration, cAMP signaling regulates surface recognition and initiation of appressorium formation in *M. oryzae*[19,48]. In pseudohyphae of *U. muhlenbergii*, individual cells are elongated and have the capsule layer, and only the cell at the tip or branching point can grow by budding, indicating the complexity of dimorphic transition. Therefore, similar to dimorphic transition in *S. cerevisiae* and *C. albicans*[49–51], the *UMP1* MAPK and cAMP-PKA pathways may coordinately regulate these processes associated with pseudohyphal/hyphal development in *U. muhlenbergii*.

Although the regulatory function of *PMK1* orthologs in fungal pathogenesis has been well characterized in plant pathogens[15,20,21], the role of this conserved MAPK pathway in fungal symbiosis is not clear because the lack of studies of its orthologs in mycorrhizae or lichen-forming fungi. In *U. muhlenbergii*, we found that *UMP1* is essential for symbiotic interactions with algal cells. The *UMP1* MAP kinase gene and its upstream MEK and MEKK genes are well-conserved in other lichen-forming ascomycetes, such as *Endocarpon pusillum*, *Acarospora strigata*, and *Cladonia grayi* that belong to Eurotiomycetes and Lecanoromycetes. Because majority of these lichen-forming fungi grow only as hyphae, the role of *UMP1* for regulating pseudohyphal growth must be unique to *U. muhlenbergii* and other dimorphic fungi. However, *UMP1* also regulates the formation of capsule layer and extracellular matrix that cover the fungal-algal cells, which is a process observed in non-dimorphic lichen-forming fungi during symbiotic interactions with algal cells in cultures or natural lichens[33,34,40]. In addition, *UMP1* is important for the differentiation of the cortex layer and entanglement of algal cells with pseudohyphae/hyphae in primitive lichen thalli of *U. muhlenbergii*. *UMP1* is also essential for the penetration of dividing algal mother cells, a process that may be common among lichen-forming fungi to avoid the aggregation of algal cells that grow by multiple fission. Therefore, Ump1 orthologs may have conserved roles in regulating the differentiation of cortex and algal layers that are associated with lichen formation in other lichen-forming fungi[4,42]. Furthermore, based on the roles of its orthologs in plant pathogens[20,21,52], the Ump1 pathway may be involved in regulating the formation of appressoria and/or haustoria as well as symbiotic growth in lichen thalli[5,26,40,45]. In this study, we observed appressorium- and haustorium-like structures in natural lichens of *U. muhlenbergii*. Therefore, as a lichen-forming fungus that is amenable to functional characterization, *U. muhlenbergii* can be used as a model to further characterize the roles of this conserved MAP kinase in fungal-algal interactions during lichen symbiosis.

The *ump1* mutant was defective in the formation of biofilm-like symbiotic complexes and differentiation of primitive lichen thalli, which may be related to its defects in pseudohyphal growth. In *M. oryzae*, because of its essential role in appressorium formation, the importance of *PMK1* for infectious growth after penetration is further characterized by inoculation through wounds or inhibition of the *PMK1*^AS mutant allele with an ATP analog 1NA-PP1[53,54]. Therefore, to dissect the functions of *UMP1* in different stages of lichen symbiosis (Fig. 6f), it may be necessary to develop additional resources or experimental approaches to assay symbiotic interactions between *U. muhlenbergii* and algal cells. Nevertheless, we were able to efficiently generate the *ump1* mutant and *ump1/UMP1*-GFP transformant in this study, further showing that *U. muhlenbergii* is amenable to molecular genetic studies. To our knowledge, the *Umgpa3*[9] and *ump1* mutants of *U. muhlenbergii* are the only two targeted gene deletion mutants generated by gene replacement in lichen forming fungi, which are under-investigated at the molecular level in general although advances in genomics and functional genomics have enabled us to address key questions related to lichen symbiosis[55]. Results from this study will be helpful to develop *U. muhlenbergii* into a model for studying molecular mechanisms regulating symbiotic interactions and lichen development.

## Methods

### Strains and culture conditions

The wild-type (JL3) and mutant strains of *U. muhlenbergii* and algal cells of *T. jamesii* were routinely cultured on potato dextrose agar (PDA) and Bold's basal medium (BBM) at 25 °C, respectively, as described previously[9]. To measure the growth rate, fungal cells were inoculated into 200 ml potato dextrose broth (PDB) at an $OD_{600}$ of 0.1 and incubated at 25 °C while shaking at 100 rpm. Samples were taken every 12 h to measure $OD_{600}$. Drops of 10 μl of fungal cells ($8 \times 10^8$/ml) were inoculated on $0.2 \times PDA$ and PDA with 0.5 M sorbitol or 5 mM IBMX (Sigma-Aldrich, St Louis, USA) and examined for yeast-to-hypha transition as described previously[9]. To assay the induction of pseudohyphal growth by algal cells, PDA plates were inoculated with drops of a mixture of fungal ($5 \times 10^4$ cells/ml) and algal ($5 \times 10^5$ cells/ml) cells, incubated at 25 °C under a 12/12 h light (1000 lux)/cycle, and examined every 24 h.

### Assays for symbiotic interactions on glass or cellulose membrane

For all the fungal-algal interaction assays, freshly harvested 10-day-old fungal and 20-day-old algal cells were mixed and cultured at 25 °C under a 12/12 h light (1000 lux)/cycle. Two approaches were tested to assay the formation of fungal-algal symbiotic complexes on glass. For the first approach, each sterile 50 mL glass flask was inoculated with 7 ml of $0.1 \times PDB$ (Becton Dickinson, Sparks Glencoe, USA) containing $8 \times 10^4$ fungal and $8 \times 10^5$ algal cells resuspended in sterile distilled water and incubated with occasional shaking for 10 days. Fungal-algal cultures were shaken gently for 1 min before examination for symbiotic complexes formed on the bottom of glass flasks. For the second test, $2 \times 10^5$ fungal and $2 \times 10^6$ algal cells were resuspended in 18 ml of $0.1 \times PDB$ and inoculated into a $\phi$9-cm petri plate with a sterile slide glass placed in the middle. After incubation for 10 days with occasional shaking, the slide glass was removed and gently rinsed with tap water for 1 min before examination. For each co-cultivation approach, five ($n = 5$) independent biological replicates were performed, with three flasks or petri plates in each replicate.

For assays with cellulose membranes, dialysis bags (Sigma-Aldrich) were boiled in 1 mM EDTA solution (pH 8.0) for 20 min, cut into single layer strips of 2 cm × 2 cm, and then placed over the surface of $0.1 \times PDA$ (Becton Dickinson) plates that had been kept in a 70 °C oven for 4 h. Drops of 105 μl $0.1 \times PDB$ containing $4 \times 10^4$ fungal and $4 \times 10^5$ algal cells were inoculated in the center of cellulose membrane strips, gently spread, and incubated for 10 days or three months before examination for fungal-algal interactions in three independent replicates ($n = 3$), with three cellulose membrane strips examined in each replicate. MEK inhibitors PD98059 and U0126 (Sigma-Aldrich)[27] were added to fungal-algal cell mixtures at the final concentration of 40 μM and 30 μM, respectively, and examined for their inhibitory effects after incubation for 10 days[9]. Each treatment was repeated three times ($n = 3$). To test the effects of rehydration on 3-month-old primitive lichen thalli formed on cellulose membranes, each co-cultivation plate was sprayed with 1 ml of sterile distilled and incubated for another week in a moisture chamber before examination. Co-cultivation plates from three ($n = 3$) independent experiments were examined for the viability of algal cells.

### Staining with DAPI, India ink, CV, and FDA and light microscope examinations

To observe the capsule layer, pseudohyphae were stained with India ink (Solarbio, Beijing, China) as described[56] and examined with a Leica DMR microscope (Wetzlar, Germany). Samples stained with 0.1% (w/v) crystal violet (Solarbio, Beijing, China) for 5 min were rinsed with 0.1 M phosphate buffer (PB, pH 7.2) for 1 min before examination as described previously[57]. Algal cells from 3-month-old primitive lichen thalli were stained with 100 μg/ml fluorescent diacetate (FDA) (Yeasen,

Wuhan, China) for 5 min and washed with 0.1 M PB buffer three times as described[58] before examination with a Leica SP8 confocal microscope (excitation at 488 nm; detection at 490–550 nm).

### SEM and TEM examinations

For SEM examination, pieces (1 mm × 1 mm) of lichen thalli and co-cultivation samples were fixed with 3% glutaradehyde in 0.1 M PB (pH 7.2) overnight at 4 °C. After rinsing with deionized water, samples were dehydrated in a series of ethanol (50, 70, 85, 95, and 100%; 15 min each), dried in a Hitachi E-1045 critical point dryer, and coated with gold particles (10–12 nm) with a Leica EM CPD300 sputter coater (Wetzlar, Germany) as described[59] before observation with a Hitachi SU8010 SEM (Tokyo, Japan) running at 5 KV. For cryo-SEM examination, lichen pieces or co-cultures on cellulose membrane were first treated in a cryo-chamber with liquid nitrogen for cryofixation for 20 s. After removing ice crystals by sublimation at −80 °C for 6 min, samples were sputter coated with gold and examined with a Hitachi S4800-II cryo SEM running at 5 KV in the frozen state as described[60].

For TEM observation, lichen pieces and co-culture samples were fixed in a mixture of 4% glutaradehyde and 2.5% formaldehyde in 0.1 PB (pH 7.2) at 4 °C for 2 h. After rinsing three times with 0.1 M PB buffer and twice with deionized water, samples were treated with 1% $KMnO_4$ at 4 °C for 1 h, rinsed three times with deionized water, and dehydrated with a series of acetone (30, 50, 70, 85, 95, and 100%; 7 min each) as described[61]. After being permeated in a series of acetone-Epon 812 mixtures (1:3 for 0.5 h, 1:1 for 1 h, and 3:1 for 1.5 h) and Epon 812 for 24 h at 60 °C, ultrathin sections (100 nm in thickness) were prepared with a Leica EM UC7 Ultramicrotome and stained with saturated uranyl acetate[62] before being examined with a JOEL JEM-1400 TEN.

### Generation of the *ump1* deletion mutant and complementation strain

To generate the *ump1* mutant with the split-marker approach[63], the flanking sequences of *UMP1* were amplified and linked to the *hph* cassette amplified from pCX63[64] by overlapping PCR with the primers listed in Supplementary Table 2. The resulting gene replacement fragments were transformed into strain JL3 by PEG-mediated protoplast transformation. Transformants resistant to 30 μg/ml hygromycin B (Sigma-Aldrich) were isolated and screened by PCR with primer pairs 5F/6R, H850/H852, 7F/HY-R, and YG-F/8R (Fig. S6b) to identify *ump1* deletion mutants.

For complementation assays, a 2574-bp fragment containing the entire *UMP1* coding region (except the TAG stop codon) and its promoter was cloned between the *Kpn*I and *Hind*III sites on pKNTG that carries the geneticin resistance marker and GFP[65] to generate the in-frame *UMP1*-GFP fusion construct pUmp1GFP. A 382-bp fragment containing the 3'-UTR and terminator sequences *UMP1* was then cloned into the *Bam*HI site on pUmp1GFP. The resulting *UMP1*-GFP construct was transformed into the *ump1* mutant as described[9]. Transformants resistant to both 30 μg/ml hygromycin B and 30 μg/ml geneticin (Sigma-Aldrich) were isolated and confirmed by PCR analysis.

### Assay for Ump1 phosphorylation

Total protein of fungal, algal, and co-cultivated fungal-algal cells were isolated with a protein lysis buffer containing the protease inhibitor cocktail (P8340) and phosphatase inhibitor cocktails 2 (P0044) and 3 (P5726) from Sigma-Aldrich as described[66]. For phosphatase treatments, 38 μl of total proteins was mixed with 800 U of λ-PPase (New England Biolabs, Ipswich, USA) and incubated at 30 °C for 30 min before being processed for loading SDS-PAGE gels as described[67]. Proteins were separated on 12% SDS-PAGE gels and transferred onto nitrocellulose membranes as described previously[28]. The expression and phosphorylation of Ump1 were detected with an anti-Gpmk1 antibody and a phosphor-p44/42 MAPK antibody (4370, Cell Signaling

Technology, Danvers, USA). The anti-Gpmk1 antibody was generated at the ABclonal Biotechnology (Wuhan, China) by injecting rabbits with a synthetic peptide of Gpmk1 of *Fusarium graminearum* (332-347 aa; DFDKHKDNLSKEQLKQ) that is highly conserved in Ump1 (332-347 aa). The 42-kD Gpmk1 band was detected in the wild type but not in the *gpmk1* mutant[68] with 1:5000 dilution of the resulting anti-Gpmk1 antibody. The expression of Ump1-GFP fusion protein was detected with an anti-GFP antibody (AF0159, Beyotime, Shanghai, China). The Pierce™ ECL western substrate kit (Thermo Fisher Scientific, Waltham, MA, USA) and Fusion-FX7 advanced imaging system (Vilber Lourmat, Eberhardzell, Germany) were used for western blot analysis. To estimate the relative phosphorylation level of Ump1, the band intensities detected by the anti-TpEY and anti-Gpmk1 antibodies was quantified with the Image-Pro Plus software. Mean and standard deviation were estimated with data from three independent replicates.

### Observation for the subcellular localization of Ump1-GFP

To examine the expression and subcellular localization of Ump1-GFP fusion proteins, freshly harvested yeast cells of the *ump1/UMP1-GFP* transformant or fungal cells co-cultivated with algal cells were examined for GFP signals with a Leica SP8 laser scanning confocal microscope (excitation at 488 nm; detection at 500-550 nm). For nucleus staining, samples were stained with 5 µg/ml 4′, 6 diamidino-2-phenylindole (DAPI) (Invitrogen) for 3 min and washed with 0.1 M PB buffer before examination by confocal microscopy (excitation at 358 nm; detection at 415–475 nm). Confocal microscopic images were analyzed for the co-localization of GFP and DAPI-staining signals with ImageJ software[69], plotting the fluorescence intensity curves of Ump1-GFP and DAPI-staining signals along the central axis of pseudohyphae.

### Phylogenetic analysis and sequence alignment with Ump1 and its orthologs

Orthologs of three predicted *U. muhlenbergii* MAPKs from S. *cerevisiae*, *M. oryzae*, and *F. graminearum* were downloaded from GenBank. Their orthologs from three lichen-forming fungi *Cladonia grayi*, *Endocarpon pusillum*, and *Acarospora strigata* were identified by BlastP searches of published genomic data[8,70–72] using homologous protein. Phylogenetic analyses were conducted with the RAxML-HPC BlackBox with 1000 bootstrap tests at the CIPRES Science Gateway website (http://www.phylo.org). Amino acid sequences were aligned with ClustalW[73] and visualized with T-coffee (http://tcoffee.crg.eu). The accession numbers and genome positions of all the MAPK proteins used for phylogenetic analysis and sequence alignment are listed in Supplementary Table S3.

### Statistics and reproducibility statement

Assays for symbiotic interactions on glass were carried out in five ($n = 5$) independent experiments. For assays with cellulose membranes, three ($n = 3$) independent experiments were conducted. For cryo-SEM examination, fifty fungal-algal complexes and three individual lichen thalli were observed in each replicate. Data from three ($n = 3$) replicates with at least 100 algal cells examined in each replicate were used to estimate the algal survival rate. The average growth rate of the wild type and *ump1* mutant was determined with data from three independent repeats ($n = 3$). For TpEY assays, the relative phosphorylation level of Ump1 in each treatment was estimated with data from three independent experiments and analyzed with the two-tailed Student's *t* test.

### Reporting summary

Further information on research design is available in the Nature Portfolio Reporting Summary linked to this article.

## Data availability

Data supporting the findings of this work are available within the paper, its Supplementary Information file, and the Source Data file. Source data are provided with this paper.

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

## Acknowledgements

We thank Ms. Jingnan Liang and Dr. Chunli Li at Institute of Microbiology, and Mr. Yanbao Tian at Institute of Genetics and Developmental Biology, CAS for assistance with TEM and SEM examinations. We also thank Drs. Guanghui Wang and Cong Jiang at Northwest A&F University for gifting the anti-Gpmk1 antibody. This work was supported by grants from National Natural Science Foundation of China (32170082) to Y.Y.W., (32070096) to X.L.W. and an AgSEED grant to J.R.X.

## Author contributions

Y.Y.W., R.L., D.W.W., B.Q. and Z.Y.B. performed the experiments and analyzed data. J.C.W. analyzed data and revised the manuscript. J.R.X., Y.Y.W. and X.L.W. designed the research and wrote the manuscript.

## Competing interests

The authors declare no competing interests.
