## [Peer Review File · Nature Communications]

Regulation of symbiotic interactions and primitive lichen
differentiation by UMP1 MAP kinase in *Umbilicaria
muhlenbergii*Reviewer #1 (Remarks to the Author):

The authors have previously provided a preliminary description of the mycobiont-photobiont interactions that underlie the lichen *Umbilicaria muhlenbergii*. Notably, the mycobiont is a dimorphic yeast that is amenable to genetic manipulation, which the authors previously exploited to document the importance of cAMP signaling in the early stages of the fungal/algal interaction. Here, the authors extend their prior work by describing the formation of differentiated primitive lichen thalli that share features with natural *U. muhlenbergii*. They describe an initial biofilm-like state that can be induced on glass surfaces. They then use extended co-incubation periods on cellulose membranes to further differentiate primitive thalli that exhibit a prominent extracellular matrix. The importance of the latter in natural lichens has been recognized. At this point, it should be noted that other lichen re-synthesis experiments have yielded generally similar results. What is notable about this study is the genetic tractability of the *U. muhlenbergii*, which allowed the authors to clearly demonstrate the necessity of the MAP kinase UMP1 for pseudohyphal growth, algal invasion, and thus formation of primitive lichen thalli. Additional results suggest that activation of UMP1 and nuclear translocation are key steps in the algal interaction. This is the first time that a clear functional genetic/genomic approach has been used to provide unambiguous demonstration of gene function in a bona fide lichen. For this reason, the results presented in the manuscript are novel and broadly significant given the growing interest in understanding the morphogenetic basis of the lichen symbiosis. While the broad applicability of these results might be questioned by some, especially given that *U. muhlenbergii* is the only known lichen with a yeast mycobiont, this study represents a noteworthy first step.

I have no serious concerns regarding the technical details of the study or the quality of the results. The following relatively minor points should be addressed by the authors;

1. line 111 and elsewhere. Was the 1:10 ratio empirically determined? How dependent are the observed interactions on the relative proportions of the partners?
2. line 228. Please provide an alignment or some other form of evidence that documents UM05c1700 as the bona fide PMK1 homologue. This should also include a representation of the number and locations of putative TEY phosphorylation sites.
3. line 243. Based on the provided image, comply,entation does not look to be complete. Is it possible to quantify pseudohyphal growth to document this?
4. line 266. Were these co-cultures on glass slides or in liquid?

Other edits;

- line 62; delete "grew on PDA".
- line 155; I'm not sure that the phrase "compromised in cell wall integrity during multiple fission" is alluding to.
- line 172; complexes, not complexed.
- line 353; it is not clear what the authors mean by "normal algal cell".
- line 701; "Crushed", not Crashed

Reviewer #2 (Remarks to the Author):

In this research the authors used a dimorphic lichen forming fungus cultivated in vitro and manipulated it genetically to show that the gene UMP1 has regulatory role and is key in the formation of the lichen symbiosis.

Working with lichen-forming fungi and algae in vitro is rather difficult due to the slow growth rate of both symbionts. I really appreciate and acknowledge the work that has been attempted in this research to seek for a regulatory gene that is implicated in the establishment of the lichen symbiosis. The authors could work with one lichen-forming fungus that can grow either as a yeast

or can generate a filamentous mycelium, and this was an advantage which eased the manipulation of the cells. But, how can we assume that this UMP1 gene acts in the same way also in other lichen-forming fungi? and in particular in those lichen-forming fungi that do have always a filamentous growth form. This is not discussed, and it is not clear if the study addresses only the particular system or could be generalized to other lichen symbioses as well.

I think that the study would profit from a better and more detailed presentation of all those studies on lichen symbioses which have dealt with symbionts isolated in vitro conditions. This because some of the morphologies that the authors report here, have been already well documented in previous works which analyzed cocultures of lichen-forming fungi and algae.

I have added further comments in the attached PDF.

The manuscript is well written, it may need further improvements from a native speaker, but there are some important flows concerning the terminology used for lichens in general, that have to be corrected.

In their review of the first version of this manuscript, reviewer #2 added some comments to the manuscript file. These comments were forwarded to the authors, who replied as included in this Peer Review File.

Reviewer #3 (Remarks to the Author):

Key results:

In this manuscript, Wang et al. characterize a gene involved in the lichen symbiosis. As their model system, the authors use the *Umbilicaria muhlenbergii* lichen – a somewhat unusual lichen symbiosis, which lends itself more easily to gene functional studies due to the dimorphic nature of its fungal symbiont (a.k.a. mycobiont). In their recent paper (Wang et al. 2020, PNAS), the authors used *Umbilicaria muhlenbergii* to conduct the first functional characterization of a gene in a lichen fungus. Here, they used the same methods to demonstrate that a MAP kinase UMP1 is required for pseudohyphal growth in the mycobiont of *Umbilicaria muhlenbergii*. Consequently, UMP1 appears necessary for the initial stages of the symbiosis development. In addition, the authors show that UMP1 phosphorylation level rises when the mycobiont is co-cultured with its algal symbiont, indicating that the MAPK pathway is triggered by the symbionts interacting with each other.

Validity:

The manuscript does not contain serious flaws but requires some additional analyses (see below).

Originality and significance:

The manuscript presents one of the first functional characterizations of a gene involved in the lichen symbiosis. These results are novel and could serve as a proof of concept for other functional studies of the lichen symbiosis. The limitations of the study are that this is an unusual lichen symbiosis that is distinct from many others, being characterized by the dimorphic nature of the mycobiont. It is therefore an open question whether the signaling pathways identified will be more broadly applicable in the symbiosis more generally. The authors should pre-emptively anticipate such criticism by modifying the discussion.

Data & methodology:

Overall, the approach and methodology are valid and provide good support to the conclusions. However, there are two areas that need to be clarified.

Many of the manuscript's conclusions are based on the analysis of images of cultures of lichen symbionts, and primitive thallus-like structures. The authors describe these results qualitatively, and do not provide cell counts to support their statements about observed phenotypes. For

example, lines 254-265 say: "When examined by SEM, most of the algal cells existed in clusters without intertwining pseudohyphae, although there were a few yeast cells on the surface in the 10-day-old ump1-algal cell masses". There really needs to be quantitative support for these assertions and statistically valid data is required. It is unacceptable to based conclusions entirely on qualitative data in this way.

Appropriate use of statistics and treatment of uncertainties:

Error bars are implied, but not defined explicitly in the legend of Fig. 1.

Conclusions:

Overall, the conclusions are supported by the data, although some additional information need to be provided in terms of quantitative support. The improvements listed below would enhance the study.

Suggested improvements:

1. More transparent and detailed descriptions of the co-culture assays and the analysis of microscopy images produced are required. Currently, the authors present the results mostly in the form of isolated images and text descriptions that sometimes use imprecise language (e.g. "most" and "a few"). These observations should be supported by cell counts and exact language regarding the observations made. In addition, the authors should clarify how many replicates were used in these experiments and that statistical support for the data presented.
2. In the phosphorylation assay (Fig. 5), the authors compared the phosphorylation level of UMP1 in lichen symbionts cultured together and separately. To be more conclusive, this analysis should include as controls the ump1 knockout mutant and ump1/UMP1 complementation strain, as well as a λ -pptase control. In addition, it would be very interesting (but not necessary) to compare the expression and phosphorylation levels of UMP1 between cultures and natural lichen thalli collected from nature. This would be a really compelling addition to the study if possible, especially given the likelihood that lichenologists will question the applicability of the findings to natural lichen thalli, as well as this system in particular.
3. The authors correctly point out that *Umbilicaria muhlenbergii* is an unusual lichen, as its mycobiont is dimorphic and undergoes dimorphic switching during the reestablishment of the symbiosis. We would be interested to hear what the authors think about how their results can be applied to other lichen symbioses. This should really be a major discussion point, given the criticisms of this system.

Minor points:

Lines 24-25: The statement that *Umbilicaria muhlenbergii* is the only lichen-forming fungus amenable to molecular manipulations seems too strong. I suggest rephrasing it to say that *Umbilicaria muhlenbergii* has proved to be amenable to molecular manipulations, or something similar.

Lines 41-42: Correct "Besides being mutually beneficial and co-dependent metabolisms" to either "Besides being mutually beneficial and co-dependent" or "Besides having mutually beneficial and co-dependent metabolic states"

Line 62: Correct "these fungal-algal cell clusters grew on PDA appeared to be" to "these fungal-algal cell clusters growing on PDA appeared to be" or "these fungal-algal cell clusters that grew on PDA appeared to be".

Line 307: Correct to "Discussion"..

Line 312: We suggest changing "old cultures" to something more specific and quantitative.

References:

The manuscript references previous literature appropriately. There are some recent lichen reviews that could be cited.

Clarity and context:

The manuscript is written clearly.

Inflammatory material:

The manuscript contains no inappropriate language and is measured in tone and conclusions.

Point-to-point responses to reviewers' comments

Reviewer #1 (Remarks to the Author):

The authors have previously provided a preliminary description of the mycobiont-photobiont interactions that underlie the lichen *Umbilicaria muhlenbergii*. Notably, the mycobiont is a dimorphic yeast that is amenable to genetic manipulation, which the authors previously exploited to document the importance of cAMP signaling in the early stages of the fungal/algal interaction. Here, the authors extend their prior work by describing the formation of differentiated primitive lichen thalli that share features with natural *U. muhlenbergii*. They describe an initial biofilm-like state that can be induced on glass surfaces. They then use extended co-incubation periods on cellulose membranes to further differentiate primitive thalli that exhibit a prominent extracellular matrix. The importance of the latter in natural lichens has been recognized. At this point, it should be noted that other lichen re-synthesis experiments have yielded generally similar results. What is notable about this study is the genetic tractability of the *U. muhlenbergii*, which allowed the authors to clearly demonstrate the necessity of the MAP kinase UMP1 for pseudohyphal growth, algal invasion, and thus formation of primitive lichen thalli. Additional results suggest that activation of UMP1 and nuclear translocation are key steps in the algal interaction. This is the first time that a clear functional genetic/genomic approach has been used to provide unambiguous demonstration of gene function in a bona fide lichen. For this reason, the results presented in the manuscript are novel and broadly significant given the growing interest in understanding the morphogenetic basis of the lichen symbiosis. While the broad applicability of these results might be questioned by some, especially given that *U. muhlenbergii* is the only known lichen with a yeast mycobiont, this study represents a noteworthy first step.

Response: Thanks for the comments.

I have no serious concerns regarding the technical details of the study or the quality of the results. The following relatively minor points should be addressed by the authors;

1. line 111 and elsewhere. Was the 1:10 ratio empirically determined? How dependent are the observed interactions on the relative proportions of the partners?

Response: Yes, the 1:10 ratio was determined experimentally in the earlier study for assaying the stimulation of pseudohyphal growth. Because the photobiont grows slower than yeast cells of the mycobiont, we tested with the ratio of 1:5, 1:10, and 1:20 and found that the ratio of 1:10 was the most suitable.

2. line 228. Please provide an alignment or some other form of evidence that documents UM05c1700 as the bona fide PMK1 homologue. This should also include a representation of the number and locations of putative TEY phosphorylation sites.

Response: The Ump1 MAP kinase is orthologous to *M. oryzae* Pmk1 and *S. cerevisiae* Fus3/Kss1. Its orthologs are well conserved among ascomycetes. In the revised manuscript, we added a supplemental figure (Fig. S4) to show that Ump1 (Um05c1700) and its orthologs from three other lichen-forming fungi are in the same cluster with Pmk1 of *M. oryzae* and Gpmk1 of *F. graminearum* in phylogenetic analysis with MAP kinases from these ascomycetes.

As suggested, we also added a supplemental figure (Fig. S5) on the sequence alignment of the regions with the TEY dual phosphorylation site to show that the T¹⁸³EY¹⁸⁵ sequence of Ump1 is well-conserved among its orthologs in kinase subdomain VIII.

3. line 243. Based on the provided image, complementation does not look to be complete. Is it possible to quantify pseudohyphal growth to document this?

Response: Fig. 4B and Fig. 4C were revised. The *ump1/UMPI* transformant was fully complemented in pseudohyphal growth and formation of primitive lichen thalli. Regarding the efficiency of pseudohyphal growth induced by algal cells (Fig. 4C), all the fungal-algal clusters examined at 10 days of co-cultivation had pseudohyphal growth in both the wild type and *ump1/UMPI* complemented transformant. No difference was observed. Related information was added in the revised manuscript.

4. line 266. Were these co-cultures on glass slides or in liquid?

Response: These were co-cultures on cellulose membranes. This sentence was revised to clarify about this point.

Other edits;

- line 62; delete "grew on PDA".

Response: Revised as suggested.

- line 155; I'm not sure that the phrase "compromised in cell wall integrity during multiple fission" is alluding to.

Response: This sentence was revised.

-line 172; complexes, not complexed.

Response: Revised as suggested.

-line 353; it is not clear what the authors mean by "normal algal cell".

Response: We deleted "normal".

-line 701; "Crushed", not Crashed

Response: Revised as suggested.

Reviewer #2 (Remarks to the Author):

In this research the authors used a dimorphic lichen forming fungus cultivated in vitro and manipulated it genetically to show that the gene UMP1 has regulatory role and is key in the formation of the lichen symbiosis.

Working with lichen-forming fungi and algae in vitro is rather difficult due to the slow growth rate of both symbionts. I really appreciate and acknowledge the work that has been attempted in this research to seek for a regulatory gene that is implicated in the establishment of the lichen symbiosis. The authors could work with one lichen-forming fungus that can grow either as a yeast or can generate a filamentous mycelium, and this was an advantage which eased the manipulation of the cells. But, how can we assume that this UMP1 gene acts in the same way also in other lichen-forming fungi? and in particular in those lichen-forming fungi that do have always a filamentous growth form. This is not discussed, and it is not clear if the study addresses only the particular system or could be generalized to other lichen symbioses as well.

I think that the study would profit from a better and more detailed presentation of all those studies on lichen symbioses which have dealt with symbionts isolated in vitro conditions. This because some of the morphologies that the authors report here, have been already well documented in previous works which analyzed cocultures of lichen-forming fungi and algae.

The manuscript is well written, it may need further improvements from a native speakers, but there are some important flows concerning the terminology used for lichens in general, that have to be corrected.

Response: Thanks for the suggestions. Regarding whether the importance of Ump1 in symbiotic interactions is applicable to other lichen-forming fungi, we added the following paragraph to the Discussion.

“The *UMPI* MAP kinase gene and its upstream MEK and MEKK genes are well-conserved in other lichen-forming ascomycetes, such as *Endocarpon pusillum*, *Acarospora strigata*, and *Cladonia grayi* that belong to Eurotiomycetes and Lecanoromycetes. Because majority of these lichen-forming fungi grow only as hyphae, the role of *UMPI* for regulating pseudohyphal growth must be unique to *U. muhlenbergii* and other dimorphic fungi. However, *UMPI* also regulates the formation of capsule layer and extracellular matrix that cover the fungal-algal cells, which is a process observed in non-dimorphic lichen-forming fungi during symbiotic interactions with algal cells in cultures or natural lichens (Honegger, 1986; Muggia, et. al., 2011; Guzow-Krzemińska and Stocker-Wörgötter, 2013). In addition, *UMPI* is important for the differentiation of the cortex layer and entanglement of algal cells with pseudohyphae/hyphae in primitive lichen thalli of *U. muhlenbergii*. *UMPI* is also essential for the penetration of dividing algal mother cells, a process that may be common among lichen-forming fungi to avoid the aggregation of algal cells that grow by multiple fission. Therefore, Ump1 orthologs

may have conserved roles in regulating the differentiation of cortex and algal layers that are associated with lichen formation in other lichen-forming fungi (Muggia, et. al., 2018; Stocker-Wörgötter and Türk, 1989). Furthermore, based on the roles of its orthologs in plant pathogens (Xu, 2000; Zhao, et. al., 2007; Jiang, et. al., 2018), the Ump1 pathway may be involved in regulating the formation of appressoria and/or haustoria as well as symbiotic growth in lichen thalli (Goodenough, et. al., 2021; Honegger, 1986; Guzow-Krzemińska and Stocker-Wörgötter, 2013). In this study, we observed appressorium- and haustorium-like structures in natural lichens of *U. muhlenbergii*. Therefore, as a lichen-forming fungus that is amenable to functional characterization, *U. muhlenbergii* can be used as a model to further characterize the roles of this conserved MAP kinase in fungal-algal interactions during lichen symbiosis.”

We also added discussions on similar differentiation processes that have been observed in earlier lichen resynthesis studies in other paragraphs in Discussions in the revised manuscript.

The revised manuscript also has incorporated your marked changes in the pdf file of this manuscript. Thanks. We appreciate that.

Line 29: any layer of cells in a lichen is not a tissue as it is a thallus. Thus it is not possible to define "epidermis" in a lichen thallus, but we can only talk about upper and lower cortex. please revise the terminology all longr the manuscript.

Response: Revised as suggested.

Line 43-45: Actually, in all truly established fungal-algal symbiosis which is a recognized lichen species, haustoria are key and are present.

Response: This sentence was revised. The word ‘some’ was deleted.

Line 113: add a fullstop after the parenthesis

Response: Revised as suggested.

Line 138: superficial

Response: Revised as suggested.

Line 144-145: This was already observed by first Honegger in the '90s and secondly by Muggia et al. in the 2010, and at that time it was called "filamentous matrix" as it breaks into filaments. This matrix can envelope the algae as well.

Response: This sentence was revised. We also added the two suggested references and related discussions in the revised manuscript.

Line 172: maybe 'complexes' or 'associations'

Response: This sentence was revised.

Line 193-195: I think this is the effect of growth towards/into the medium to search for nutrients, as it often happens that the mycelia enter the medium and grow inside

Response: Thanks for the comment. If it is simply for nutrient uptake, there is no need to form a focal point on a permeable membrane. It appears to be a developmental process and it is only formed by relatively old co-cultures. However, the suggested explanation is also possible. Therefore, discussion revised to include both possibilities.

Line 701: squashed

Response: This sentence was revised.

Reviewer #3 (Remarks to the Author):

Key results:

In this manuscript, Wang et al. characterize a gene involved in the lichen symbiosis. As their model system, the authors use the *Umbilicaria muhlenbergii* lichen – a somewhat unusual lichen symbiosis, which lends itself more easily to gene functional studies due to the dimorphic nature of its fungal symbiont (a.k.a. mycobiont). In their recent paper (Wang et al. 2020, PNAS), the authors used *Umbilicaria muhlenbergii* to conduct the first functional characterization of a gene in a lichen fungus. Here, they used the same methods to demonstrate that a MAP kinase UMP1 is required for pseudohyphal growth in the mycobiont of *Umbilicaria muhlenbergii*. Consequently, UMP1 appears necessary for the initial stages of the symbiosis development. In addition, the authors show that UMP1 phosphorylation level rises when the mycobiont is co-cultured with its algal symbiont, indicating that the MAPK pathway is triggered by the symbionts interacting with each other.

Validity:

The manuscript does not contain serious flaws but requires some additional analyses (see below).

Originality and significance:

The manuscript presents one of the first functional characterizations of a gene involved in the lichen symbiosis. These results are novel and could serve as a proof of concept for other functional studies of the lichen symbiosis. The limitations of the study are that this is an unusual lichen symbiosis that is distinct from many others, being characterized by the dimorphic nature of the mycobiont. It is therefore an open question whether the signaling pathways identified will be more broadly applicable in the symbiosis more generally. The authors should pre-emptively anticipate such criticism by modifying the discussion.

Response: The following paragraph was added in Discussion to discuss the importance of this well-conserved MAP kinase pathway in symbiotic interactions in lichen-forming fungi.

“The *UMPI* MAP kinase gene and its upstream MEK and MEKK genes are well-conserved in other lichen-forming ascomycetes, such as *Endocarpon pusillum*, *Acarospora strigata*, and *Cladonia grayi* that belong to Eurotiomycetes and Lecanoromycetes. Because majority of these lichen-forming fungi grow only as hyphae, the role of *UMPI* for regulating pseudohyphal growth must be unique to *U. muhlenbergii* and other dimorphic fungi. However, *UMPI* also regulates the formation of capsule layer and extracellular matrix that cover the fungal-algal cells, which is a process observed in non-dimorphic lichen-forming fungi during symbiotic interactions with algal cells in cultures or natural lichens (Honegger, 1986; Muggia, et. al., 2011; Guzow-Krzemińska and Stocker-Wörgötter, 2013). In addition, *UMPI* is important for the differentiation of the cortex layer and entanglement of algal cells with pseudohyphae/hyphae in primitive lichen thalli of *U. muhlenbergii*. *UMPI* is also essential for the penetration of dividing algal mother cells, a process that may be common among lichen-forming fungi to avoid the aggregation of algal cells that grow by multiple fission. Therefore, Ump1 orthologs may have conserved roles in regulating the differentiation of cortex and algal layers that are associated with lichen formation in other lichen-forming fungi (Muggia, et. al., 2018; Stocker-Wörgötter and Türk, 1989). Furthermore, based on the roles of its orthologs in plant pathogens (Xu, 2000; Zhao, et. al., 2007; Jiang, et. al., 2018), the Ump1 pathway may be involved in regulating the formation of appressoria and/or haustoria as well as symbiotic growth in lichen thalli (Goodenough, et. al., 2021; Honegger, 1986; Guzow-Krzemińska and Stocker-Wörgötter, 2013). In this study, we observed appressorium- and haustorium-like structures in natural lichens of *U. muhlenbergii*. Therefore, as a lichen-forming fungus that is amenable to functional characterization, *U. muhlenbergii* can be used as a model to further characterize the roles of this conserved MAP kinase in fungal-algal interactions during lichen symbiosis.”

Data & methodology:

Overall, the approach and methodology are valid and provide good support to the conclusions. However, there are two areas that need to be clarified.

Many of the manuscript's conclusions are based on the analysis of images of cultures of lichen symbionts, and primitive thallus-like structures. The authors describe these results qualitatively, and do not provide cell counts to support their statements about observed phenotypes. For example, lines 254-265 say: “When examined by SEM, most of the algal cells existed in clusters without intertwining pseudohyphae, although there were a few yeast cells on the surface in the 10-day-old ump1-algal cell masses”. There really needs

to be quantitative support for these assertions and statistically valid data is required. It is unacceptable to based conclusions entirely on qualitative data in this way.

Response: In the revised manuscript, quantitative data were added for fungal-algal cell masses attached to the slide glass, extracellular matrix produced by fungal-algal complexes formed on cellulose membranes, hairy appearance of 3-month symbiotic complexes, and the survival rate of algal cells.

Appropriate use of statistics and treatment of uncertainties:

Error bars are implied, but not defined explicitly in the legend of Fig. 1.

Response: Related figure legend was revised to define the error bars in Figure 5. (Fig. 5, but not Fig. 1, has error bars in the figure).

Conclusions:

Overall, the conclusions are supported by the data, although some additional information need to be provided in terms of quantitative support. The improvements listed below would enhance the study.

Suggested improvements:

1. More transparent and detailed descriptions of the co-culture assays and the analysis of microscopy images produced are required. Currently, the authors present the results mostly in the form of isolated images and text descriptions that sometimes use imprecise language (e.g. “most” and “a few”). These observations should be supported by cell counts and exact language regarding the observations made. In addition, the authors should clarify how many replicates were used in these experiments and that statistical support for the data presented.

Response: Revised as suggested. In the revised manuscript, we added more details to Materials and Methods, and presented quantitative data of fungal-algal cell masses attached to the slide glass, extracellular matrix produced by fungal-algal complexes formed on cellulose membranes, hairy appearance of 3-month symbiotic complexes, and the survival rate of algal cells.

2. In the phosphorylation assay (Fig. 5), the authors compared the phosphorylation level of UMP1 in lichen symbionts cultured together and separately. To be more conclusive, this analysis should include as controls the *ump1* knockout mutant and *ump1/UMP1* complementation strain, as well as a λ -pptase control. In addition, it would be very interesting (but not necessary) to compare the expression and phosphorylation levels of UMP1 between cultures and natural lichen thalli collected from nature. This would be a really compelling addition to the study if possible, especially given the likelihood that lichenologists will question the applicability of the findings to natural lichen thalli, as well as this system in particular.

Response: During revision, we repeated TEY assays as suggested. Figure 5 was revised to present results from TEY assays with suggested controls. Because the UmMps1 band is stronger than the Ump1 band, we add TEY assays with the *ump1/UMPI-GFP* transformant in which the 69-Ump1-GFP band was detected. Regarding natural lichen thalli, thanks for the suggestion. Our preliminary data showed that the phosphorylation level of Ump1 (42-kD band) was similar in natural lichens and fungal-algal co-cultures. However, the phosphorylation of UmMps1 (48-kD band) was barely detectable although its expression was normal in lichen thalli harvested from rock surface, which is different from fungal-algal co-cultures. This is something intriguing and will need further tests. Currently, one student is working on UmMps1 and we wish to present related data in a separate manuscript focusing on UmMps1. Again, thanks for the suggestion.

3. The authors correctly point out that *Umbilicaria muhlenbergii* is an unusual lichen, as its mycobiont is dimorphic and undergoes dimorphic switching during the reestablishment of the symbiosis. We would be interested to hear what the authors think about how their results can be applied to other lichen symbioses. This should really be a major discussion point, given the criticisms of this system.

Response: This point is similar to the comment on Originality and significance above. We added one paragraph in Discussion (see above) to discuss the importance of this well-conserved MAP kinase pathway in symbiotic interactions in lichen-forming fungi.

Minor points:

Lines 24-25: The statement that *Umbilicaria muhlenbergii* is the only lichen-forming fungus amenable to molecular manipulations seems too strong. I suggest rephrasing it to say that *Umbilicaria muhlenbergii* has proved to be amenable to molecular manipulations, or something similar.

Response: Revised as suggested.

Lines 41-42: Correct “Besides being mutually beneficial and co-dependent metabolisms” to either “Besides being mutually beneficial and co-dependent” or “Besides having mutually beneficial and co-dependent metabolic states”

Response: Revised as suggested.

Line 62: Correct “these fungal-algal cell clusters grew on PDA appeared to be” to “these fungal-algal cell clusters growing on PDA appeared to be” or “these fungal-algal cell clusters that grew on PDA appeared to be”.

Response: We deleted “grew on PDA”.

Line 307: Correct to “Discussion”.

Response: Revised as suggested.

Line 312: We suggest changing “old cultures” to something more specific and quantitative.

Response: This sentence was revised.

References:

The manuscript references previous literature appropriately. There are some recent lichen reviews that could be cited.

Response: Revised as suggested. Two most recent reviews were added in the revised manuscript. (Scharnagl, et. al., 2023. The coming golden age for lichen biology. Pichler, et. al., 2023. How to build a lichen: from metabolite release to symbiotic interplay).

Reviewer #3 (Remarks to the Author):

I am grateful to the authors for responding so constructively to my comments. The paper is much improved and makes a valuable contribution. My only residual concerns are that I would still appreciate it if the authors could show more detail of quantification as this adds evidence for their major conclusions.

Specifically:

Page 5, line 121: how many mycobiont cells were examined? Did they come from one and the same co-cultivation plate?

Page 6, line 137: how many such complexes were examined?

Page 16, line 445: how many flasks have been used in the co-culture experiment? Were the multiple replicates?

Page 16, lines 454-464: how many co-cultivation plates were made?

Please can you address these final issues in your revision, as it adds essential information in my view for readers.

Point-to-point responses to reviewers' comments

Reviewer #3 (Remarks to the Author):

I am grateful to the authors for responding so constructively to my comments. The paper is much improved and makes a valuable contribution. My only residual concerns are that I would still appreciate it if the authors could show more detail of quantification as this adds evidence for their major conclusions.

Response: Thanks for the comments. We added more detailed quantitative information in revised manuscript.

Specifically:

Page 5, line 121: how many mycobiont cells were examined? Did they come from one and the same co-cultivation plate?

Response: Five independent replicates were performed to observe the fungal-algal interactions. Each replicate had three culture plates and 100 fungal cells were observed in each plate.

Page 6, line 137: how many such complexes were examined?

Response: A total of 50 fungal-algal complexes were observed for extracellular matrix under cryo-SEM.

Page 16, line 445: how many flasks have been used in the co-culture experiment? Were the multiple replicates?

Response: A total of 15 flasks were observed for co-culture phenotypes from five independent replicates.

Page 16, lines 454-464: how many co-cultivation plates were made?

Response: This experiment involved three independent replicates. Each replicate had three culture plates.

Please can you address these final issues in your revision, as it adds essential information in my view for readers.

Response: The details of the above-mentioned replicates and quantitative information were added to the revised manuscript.